

# The associations between well-being and Cloninger's personality dimensions in a Korean community sample

Soo Jin Lee[1], C. Robert Cloninger[2,3] and Han Chae[4]

[1] Department of Psychology, Kyungsung University, Busan, Republic of Korea
[2] Department of Psychiatry, Washington University in St. Louis, St. Louis, MO, United States of America
[3] Anthropedia Foundation, St. Louis, MO, United States of America
[4] School of Korean Medicine, Pusan National University, Busan, Republic of Korea

## ABSTRACT

**Background**. Well-being is a multidimensional construct comprising affective and non-affective components. Previous research has consistently linked personality traits to well-being, yet cultural variations in this association remain underexplored, particularly in collectivistic cultures such as Korea. Therefore, this study aims to identify universal and culture-specific characteristics of personality in relation to well-being.

**Methods**. A sample of 527 Korean university students participated, providing data through the Korean version of the Temperament and Character Inventory-RS (TCI-RS), self-rated health (SRH), social support (SS), Satisfaction With Life Scale (SWLS), and positive and negative affect schedule (PANAS). Pearson correlation analysis and ANCOVA, with sex and age as covariates, were employed to examine linear associations. Multidimensional personality profiles were utilized to investigate non-linear associations among character dimensions on different aspects of well-being. All analysis was performed using jamovi 2.3.12.

**Results**. Self-directedness and cooperativeness exhibited positive linear associations with both affective (positive and negative affect) and non-affective (SRH, SS, SWLS) components of well-being. Self-directedness emerged as a key predictor across various well-being aspects. Cooperativeness was strongly associated with perception of social support. Self-transcendence showed positive associations with both positive and negative affect, considering interactions with other character dimensions.

**Discussion**. While self-directedness played a pivotal role universally, the impact of cooperativeness and self-transcendence appeared to be influenced by cultural factors, enhancing perception of social support and affecting both positive and negative affect in a collectivistic culture. This study illustrates the importance of considering cultural nuances in the relationship between personality and well-being. Future research should delve deeper into cultural differences, emphasizing the need for subtle interpretations of specific personality traits within diverse cultural contexts.

Corresponding author
Han Chae, han@chaelab.org

## INTRODUCTION

Personality, as defined by *Cloninger (2004)*, is a dynamic psychobiological internal system, comprising temperament and character dimensions. It serves as the mechanism through which individuals shape and adapt to both internal and external changes. Temperament is considered to be a key element as it is connected to reacting to emotional stimuli, variations in the intensity of responses to emotional events, and the length of emotional reactions (*Kim-Prieto et al., 2005*; *Nigg, 2006*).

The four temperament dimensions are defined in terms of individual differences in behavioral learning mechanisms, explaining responses to novelty and signals of reward or relief of punishment (Novelty Seeking: NS), responses to signals of punishment or non-reward (Harm Avoidance: HA), responses to social and attachment rewards (Reward Dependence: RD), and the maintained response to previously rewarded behavior with intermittent reinforcement (Persistence: PS) (*Cloninger, 1987*; *Cloninger, Svrakic & Przybeck, 1993*).

In contrast, character refers to an individual's ability to accept oneself, align behavior with personal goals, demonstrate empathy, assist others, and foster a sense of self-efficacy, autonomy, and responsibility (*Cloninger, Svrakic & Przybeck, 1993*). Character is composed of three dimensions: Self-Directedness (SD) (based on the concept of the self as an autonomous individual) enables people to participate in purposeful actions; Cooperativeness (CO) (based on the concept of the social self) enables people to be tolerant and flexible about choices regarding goals because thought and behavior are based on mutual interests with other persons; and Self-Transcendence (ST) (based on the concept of the self with values derived from awareness of being an integral aspect of a larger whole) enables people to intuitively recognize the values and meaning in all things.

In summary, character empowers intentional actions and the interpretation of our experiences, facilitating the self-regulation of emotional reactions and even our habits (*Cloninger, 2004*; *Fahlgren et al., 2015*; *Moreira, Inman & Cloninger, 2021*). Due to its distinction between nonintentional (*i.e.,* temperament) and intentional (*i.e.,* character) domains of personality, Cloninger's biopsychosocial model is appropriate for assessment of both within-person learning processes and between-person differences (*Cervone, 2005*; *Lee, Jeong & Chae, 2023*)—that is, the way people differ from others but also the processes that motivate and regulate adaptive processes occurring within the individual.

Well-being encompasses the affective and non-affective dimensions of an individual's subjective experience resulting from the personal assessment of various life dimensions. A high level of well-being should not be equated with a life devoid of challenges or solely characterized by positive events. Individuals can adapt to challenges, evaluating their lives in light of changing circumstances (*Diener et al., 1999*; *Lent, 2004*; *McDowell, 2010*). Well-being is often conceptualized as hedonic and/or eudaimonic well-being. Hedonic well-being indicates how and why individuals think and feel their lives in positive ways, and consists of a combination of negative and positive emotions and life satisfaction (*Diener, 1984*). Eudaimonic well-being encompasses the wider domains of personal growth, purposeful engagement and self development (*Henderson & Knight, 2012*; *Ryff,*

*Singer & Dienberg Love, 2004*). The notions of hedonic and eudaimonic well-being are separate yet interconnected facets of psychological functioning, and a comprehensive understanding of well-being necessitates both (*Carver & Connor-Smith, 2010*; *Keyes, Shmotkin & Ryff, 2002*). Eudaimonia, often referred to as psychological well-being has been employed to describe well-being that emerges from a combination of character strengths encompassing aspects of SD (*e.g.*, autonomy, life purpose, environmental mastery, and self-acceptance), CO (*e.g.*, positive relations with others), and ST (*e.g.*, personal growth and self-actualization) (*Ryan & Deci, 2001*; *Ryff & Keyes, 1995*; *Schmutte & Ryff, 1997*). That is, character dimensions aim at illustrating maturity and integration of personality.

There is some literature regarding the associations between well-being and character. First, in an Israeli population-based study, researchers (*Cloninger & Zohar, 2011*) used a person-centered approach with multidimensional personality profiles. They investigated how personality profiles influence physical, emotional, and social aspects of well-being. They concentrated on character traits assessed through the Temperament and Character Inventory (TCI) (*Cloninger, Svrakic & Przybeck, 1993*). SD was strongly linked to various aspects of well-being, including life satisfaction, social support, subjective health, positive affect, and negative affect. CO was associated especially with perceived social support, and ST predicted positive emotions when accounting for the influence of the other character dimensions. The research demonstrated that personality traits exert significant influence on the perception of well-being.

Second, Josefsson and his colleagues (*Josefsson et al., 2011*) also confirmed that SD was strongly associated with both affective (positive and negative affect) and non-affective (life satisfaction, perceived social support, or subjective health) well-being using a Finnish sample. They found that CO had a stronger association with perceived social support when accounting for the influence of the other character dimensions. In addition, they reported that ST showed both positive and negative association sometimes referred to as a "twin effect" when the influence of the other character dimensions was taken into account. This study suggests that character plays an important role in the perception of well-being depending on the socio-cultural atmosphere.

Third, *Moreira et al. (2015)* also found that SD exhibited a robust association with both affective and non-affective well-being using Portuguese adolescents. However, they found that social support and positive affect were not associated with CO and ST, respectively. Furthermore, these associations were repeatedly found in a subsequent study in *Moreira, Inman & Cloninger (2023)* while they did not measure subjective health and perceived social support in relation to well-being.

Fourth, *Giakoumaki et al. (2016)* reported that groups with higher scores of SD (SCT, SCt, ScT, Sct) character profiles showed higher scores of positive affect and lower scores of negative affect than those in groups with lower scores of SD (sct, scT, sCt, sCT) profiles. They used a sample of 480 Greek community members with a higher education level. That is, Self-directedness was strongly associated with both positive and negative affect. They did not observe the non-affective aspect of well-being.

Summing up, in terms of non-linear associations, life satisfaction has a positive association with SD but not with CO or ST. Perceived social support has also a positive

association with SD and CO but not with ST. Subjective health has a positive association with SD but not with CO and ST. The positive association between positive affect and three character dimensions (SD, CO, ST) and negative association between negative affect and three character dimensions were repeatedly reported. However, inconsistencies were reported regarding positive association between negative affect and ST in some previous studies (*Josefsson et al., 2011*; *Moreira, Inman & Cloninger, 2023*).

While there are consistent but somewhat mixed associations between well-being and character, the present study seeks to explain this in light of socio-cultural aspects. In addition, cultural influences, along with factors such as social norms and the perceived importance of individual well-being, are said to impact perceptions of well-being (*Diener, Oishi & Lucas, 2003*). Given that well-being may be evaluated with different criteria in different cultures (*Heine et al., 2002*), it is crucial to consider such variability when evaluating the universal relationship between personality and well-being. Therefore, the current study aims to examine if these cross-cultural variations impact the association between personality and well-being.

For this purpose, we assess how specific combinations of character traits relate to both affective and non-affective dimensions of well-being in a population-based Korean sample, similar to the previous studies (*Cloninger & Zohar, 2011*; *Giakoumaki et al., 2016*; *Josefsson et al., 2011*; *Moreira et al., 2015*; *Moreira, Inman & Cloninger, 2023*). That is, non-linear associations, a person-centered approach to personality is a key component in our study. An approach that conceptualizes personality as a blend of multiple components, rather than individual dimensions examined in isolation, allows for a comprehensive understanding of processes within individuals. It goes beyond merely highlighting differences between individuals, considering the complex biopsychosocial reality they encounter (*Bergman & Magnusson, 1997*). We also add analysis of linear association to non-linear methods to address the complexity of developmental processes. Therefore, we propose the following hypotheses: (1) SD will have a strong association with both affective and non-affective aspects of well-being, (2) CO will provide additional insight into aspects of social support, and (3) ST will be associated with affective aspects of well-being.

To the best of our knowledge, the present study is the first to investigate well-being using TCI in the East Asian cultural context. Considering Eastern societies as collectivist and Western societies as individualist cultural contexts (*Barkema et al., 2015*), we aimed to explore whether the associations between personality and well-being would appear similarly in Western and Eastern cultural context. Through this, we intend to examine how the sociocultural norms that are implicitly imposed on us through cultural differences could potentially influence our perspective on well-being.

## METHODS

### Subjects

TCI-Revised-Short Version (TCI-RS), Self-Rated Health (SRH), Social Support (SS), Satisfaction With Life Scale (SWLS), and Positive Affect Negative Affect Schedule (PANAS) were used to measure temperament and character, perceived health and social support,

life satisfaction, and positive and negative affect, respectively, in a sample of 527 university students (195 males and 332 females) in the Busan Metropolitan area, South Korea. Out of 600 students, 527 (87.83%) volunteered to participate and completed our questionnaires. Participants were also asked about their age and gender. This study was approved by the Institutional Ethics Board of Kyungsung University, South Korea (approval number KSU-19-03-005) and all participants provided written informed consent.

## Measures

### Temperament and character inventory

The Korean TCI-Revised Short (TCI-RS) has 140 items with a 5-point Likert scale ranging from not at all (0) to very true (4). The items measure four traits of temperament and three traits of character. The current study included only the character traits. The character dimension has three subscales of Self-Directedness (SD; characterized as purposeful, resourceful, responsible, and self-accepting), Cooperativeness (CO; empathic, forgiving, helpful, and tolerant), and Self-Transcendence (ST; contemplating, spiritual, idealistic, and transpersonal) (*Cloninger, Svrakic & Przybeck, 1993*; *Lee et al., 2014*; *Min, Oh & Lee, 2007*). The internal consistencies of NS, HA, RD, PS, SD, CO, and ST were reported to be .83, .86, .81, .82, .87, .76, and .90, respectively (*Min, Oh & Lee, 2007*). The internal consistencies of NS, HA, RD, PS, SD, CO, and ST in this study were .82, .62, .53, .77, .73, .70, and .89, respectively.

### Well-being measures

**Self-Rated Health (SRH)** measures subjective health asking ''what is your health like compared to others of your age?'' (*Choi, 2016*). The SRH has one item with a 5-point Likert scale ranging from very bad (1) to very good (5).

   **Social Support (SS)** was used to measure social support which was made by *Park (1985)* and then revised by *Kim (1995)*. The SS includes 25 items with a 5-point Likert scale ranging from not at all (1) to very true (5). It has four subscales of emotional support, informational support, material support and assessment support. The total score of the SS ranges from 25 to 125, with higher scores indicating higher social support. The internal consistency of SS was .94 (*Park, 1985*). The internal consistency in this study was .96.

   **Satisfaction with Life Scale (SWLS)** was used to assess life satisfaction (*Diener et al., 1985*). The SWLS includes 5 items with a 7-point Likert scale ranging from not at all (1) to extremely true (7). The total score of the SWLS ranges from 5 to 35, with higher scores indicating higher life satisfaction. We used the version of SWLS translated into Korean by *Lee (2004)* and the internal consistency of SWLS was .85 (*Lee, 2004*). The internal consistency in this study was .86.

   **Positive Affect Negative Affect Schedule (PANAS)** was employed to assess positive and negative affect (*Watson, Clark & Tellegen, 1988*). The PANAS includes 10 items of positive affect and 10 items with a 5-point Likert scale ranging from not at all (1) to very true (5). The total score of the PANAS ranges from 10 to 50, with higher scores indicating higher positive or negative affect, respectively. We used the version of PANAS translated into Korean by Park and Lee and the internal consistency of PANAS was .81 (*Park & Lee, 2016*). The internal consistency in this study was .82.

**Composite Health Index (CHI)** was employed as a standardized summary measure of perceived non-affective wellness of subjective health, social support, and life satisfaction (*Cloninger & Zohar, 2011*). The original scores of well-being measures (SRH, SS, SWLS, and PANAS) were transformed (mean = 1, SD = 1) into standardized scores of SRHz, SSz, SWLSz, PAz, and NAz to facilitate comparison and informative illustrations of differences between character groups. The CHI was the mean of three non-affective well-being measures of SRHz, SSz, and SWLSz.

## Statistical analysis

All data underwent meticulous verification for potential miscoding, assessment of value distribution, and correction of missing values prior to analysis. Then, significant differences between male and female participants in age, character, and well-being (SRH, SS, SWLS, PA, and NA) were examined using t-tests. Pairwise correlations between character and well-being were determined using Pearson's correlation, and correlation coefficients were provided.

To examine the non-linear associations between personality configurations and well-being, character profiles were defined. The participants were grouped according to all the possible combinations of high and low scores in each one of the character dimensions (Table 1). Our non-linear analyses were based on Cloninger's proposal for character profiles (*Cloninger, 2004*; *Cloninger & Zohar, 2011*; *Josefsson et al., 2011*). We conducted an analysis of well-being measures for different character profiles to examine the differences in estimated well-being measures, with gender and age controlled, using ANCOVA. In cases where the F-value was statistically significant, the Bonferroni post-hoc test was utilized for further explorations. Profile-based configural analysis of the influence of each character trait on both non-affective and affective components of well-being was conducted to reveal more information than ANCOVA.

The data were presented as means and standard deviations or frequency, and estimated values considering gender and age were illustrated as means and standard errors. All analyses were performed using jamove 2.3.12 (The jamovi Team) (*The Jamovi Project, 2022*; *R Core Team, 2021*).

# RESULTS

## Demographic features

The demographic features of age, character subscales, SRH, SS, SWLS, PA, and NA were described in Table 2. There were significant differences in age ($t = 4.276$, $p < 0.001$) between males and females.

In the current study, female students were found to be significantly high in ST (20.92 ± 11.30 for males and 24.26 ± 11.59 for females, respectively), and significantly low in SD (45.34 ± 11.51, 41.91 ± 12.16). Male students were found to have significantly lower scores in SS (92.83 ± 14.69, 98.22 ± 14.17) and NA (18.91 ± 7.27, 20.55 ± 8.57), and significantly higher in SRH (3.75 ± 0.94, 3.37 ± 1.00), SWLS (21.89 ± 5.85, 20.59 ± 6.34), and PA (29.06 ± 7.55, 26.71 ± 7.61).

**Table 1** Frequency distribution of TCI character profiles.

| Character profile | N (women/men) | Percentage (women/men) |
|---|---|---|
| SCT—creative | 71 (49/22) | 13.5 (69.0/31.0) |
| SCt—organized | 95 (44/51) | 18.0 (46.3/53.7) |
| ScT—fanatical | 26 (16/10) | 4.9 (61.5/38.5) |
| Sct—autocratic | 58 (35/23) | 11.0 (60.3/39.7) |
| sCT—moody | 42 (34/8) | 8.0 (81.0/19.0) |
| sCt—dependent | 70 (50/20) | 13.3 (71.4/28.6) |
| scT—disorganized | 74 (45/29) | 14.0 (60.8/39.2) |
| sct—depressive | 91 (59/32) | 17.3 (64.8/35.2) |
| Total | 527 (332/195) | 100 (63.0/37.0) |

**Table 2** Demographic features of the current study.

| | Males ($n = 195$) | Females ($n = 332$) | Statistics |
|---|---|---|---|
| Age[***] | $21.02 \pm 2.19$ | $20.21 \pm 2.05$ | $t = 4.276, p < 0.001$ |
| Character | | | |
| SD[**] | $45.34 \pm 11.51$ | $41.91 \pm 12.16$ | $t = 3.192, p = 0.002$ |
| CO | $56.95 \pm 10.91$ | $56.80 \pm 9.92$ | $t = 0.171, p = 0.864$ |
| ST[**] | $20.92 \pm 11.30$ | $24.26 \pm 11.59$ | $t = -3.227, p = 0.001$ |
| Well-being | | | |
| SRH[***] | $3.75 \pm 0.94$ | $3.37 \pm 1.00$ | $t = 4.32, p < 0.001$ |
| SS[***] | $92.83 \pm 14.69$ | $98.22 \pm 14.17$ | $t = -4.165, p < 0.001$ |
| SWLS[*] | $21.89 \pm 5.85$ | $20.59 \pm 6.34$ | $t = 2.335, p = 0.020$ |
| PA[***] | $29.06 \pm 7.55$ | $26.71 \pm 7.61$ | $t = 3.43, p = 0.001$ |
| NA[*] | $18.91 \pm 7.27$ | $20.55 \pm 8.57$ | $t = -2.327, p = 0.020$ |

Notes.

[*]$p < .05$
[**]$p < .01$
[***]$p < .001$

SD, Self-Directedness; CO, Cooperativeness; ST, Self-Transcendence; SRH, Self-Rated Health; SS, Social Support; SWLS, Satisfaction with Life Scale; PA, Positive Affect; NA, Negative Affect.

## Correlation coefficients among measures

The correlation coefficients among TCI subscales and well-being measures was shown in Table 3. The SRH revealed significant positive correlation with SD ($r = 0.471, p < 0.001$) and CO ($r = 0.170, p < 0.001$). The SS was significantly correlated positively with SD ($r = 0.338, p < 0.001$) and CO ($r = 0.406, p < 0.001$). The SWLS revealed significant positive correlation with SD ($r = 0.599, p < 0.001$), CO ($r = 0.272, p < 0.001$) and ST ($r = 0.100, p < 0.05$). The CHI was correlated positively with SD ($r = 0.627, p < 0.001$), CO ($r = 0.377, p < 0.001$), SRH ($r = 0.731, p < 0.001$), SS ($r = 0.728, p < 0.001$), SWLS ($r = 0.788, p < 0.001$) and PA ($r = 0.476, p < 0.001$), and negatively with NA ($r = -0.363, p < 0.001$).

The PA was significantly correlated positively with SD ($r = 0.419, p < 0.001$), CO ($r = 0.265, p < 0.001$), and ST ($r = 0.224, p < 0.001$). The NA was significantly correlated

**Table 3** Correlation coefficients among character subscales and well-being measures.

| | SD | CO | ST | SRH | SS | SWLS | CHI | PA |
|---|---|---|---|---|---|---|---|---|
| CO | **0.325**[***] | | | | | | | |
| ST | −0.119[**] | 0.041 | | | | | | |
| SRH | **0.471**[***] | 0.17[***] | −0.061 | | | | | |
| SS | **0.338**[***] | **0.406**[***] | 0.072 | 0.254[***] | | | | |
| SWLS | **0.599**[***] | 0.272[***] | 0.100[*] | **0.389**[***] | **0.382**[***] | | | |
| CHI | **0.627**[***] | **0.377**[***] | 0.049 | **0.731**[***] | **0.728**[***] | **0.788**[***] | | |
| PA | **0.419**[***] | 0.265[***] | 0.224[***] | **0.344**[***] | 0.276[***] | **0.449**[***] | **0.476**[***] | |
| NA | **−0.437**[***] | −0.146[***] | 0.294[***] | **−0.381**[***] | −0.173[***] | −0.261[***] | **−0.363**[***] | −0.012 |

**Notes.**
[*]$p < .05$
[**]$p < .01$
[***]$p < .001$
Bold represents coefficient larger than 0.3.

SD, Self-Directedness; CO, Cooperativeness; ST, Self-Transcendence; SRH, Self-Rated Health; SS, Social Support; SWLS, Satisfaction with Life Scale; CHI, Composite Health Index; PA, Positive Affect; NA, Negative Affect.

negatively with SD ($r = -0.437, p < 0.001$), CO ($r = -0.146, p < 0.001$), and ST ($r = -0.294, p < 0.001$).

### Differences of well-being score of character profiles

We examined the well-being measures (SRH, SS, SWLS, PA, and NA) for eight groups based on the character profiles (SCT, SCt, ScT, Sct, sCT, sCt, scT, and sct). The comparison of estimated well-being measures with raw scores was presented in Table 4.

The standardized scores of non-affective well-being (SRHz, SSz, and SWLSz) of each character group were presented in Fig. 1. The composite well-being index of each character group was in decreasing order as illustrated in Fig. 2A. The top sixth and the bottom sixth of the distribution of the CHI were selected and labeled as "best health" and "worst ill-health" to reveal the adaptive functions. The proportion of extremely good health and extremely poor health in each character group were shown in Fig. 2B. The standardized score of affective well-being (PAz and NAz) of each character group was presented in Fig. 3.

### Comparison results of well-being score of character profiles

Taking interactions among the character traits into account, higher Self-directedness was associated with greater self-rated health and life satisfaction in the non-affective components as well as with positive and negative affect in affective components of well-being (Table 5). Higher Cooperativeness was strongly associated with greater social support in all contrasts, but had little or no association with other affective and non-affective components of well-being (Table 5). Self-transcendence had little or no association with any measure of affective and non-affective components of well-being (Table 5).

## DISCUSSION

The study's findings closely align with previous research (*Cloninger & Zohar, 2011*; *Giakoumaki et al., 2016*; *Josefsson et al., 2011*; *Moreira et al., 2015*; *Moreira, Inman &*

Lee et al. (2024), *PeerJ*, DOI 10.7717/peerj.18379

**Table 4  Estimated well-being measures of character profile groups.**

|  | SCT | SCt | ScT | Sct | sCT | sCt | scT | sct | Statistics |
|---|---|---|---|---|---|---|---|---|---|
| *n* | 71 | 95 | 26 | 58 | 42 | 70 | 74 | 91 | |
| SRH | $3.99 \pm 0.11$ | $3.91 \pm 0.09$ | $4.00 \pm 0.18$ | $3.64 \pm 0.12$ | $3.31 \pm 0.14$ | $3.19 \pm 0.11$ | $3.11 \pm 0.11$ | $3.20 \pm 0.10$ | $F = 10.618$, $p < 0.001$ (SCT>sCT, sCt, scT and sct, SCt>sCt, scT and sct, ScT>sCt, scT and sct, Sct>scT) |
| SS | $104.04 \pm 1.57$ | $101.14 \pm 1.35$ | $94.84 \pm 2.59$ | $94.14 \pm 1.74$ | $100.96 \pm 2.04$ | $98.69 \pm 1.58$ | $89.97 \pm 1.54$ | $87.73 \pm 1.38$ | $F = 13.906$, $p < 0.001$ (SCT>Sct, scT and sct, SCt>Sct, scT and sct, sCT>scT and sct, sCt>scT and sct) |
| SWLS | $25.59 \pm 0.64$ | $23.90 \pm 0.55$ | $24.19 \pm 1.06$ | $21.92 \pm 0.71$ | $19.33 \pm 0.83$ | $18.77 \pm 0.64$ | $20.11 \pm 0.63$ | $16.53 \pm 0.56$ | $F = 20.22$, $p < 0.001$ (SCT>Sct, sCT, sCt, scT, and sct, SCt>sCT, sCt, scT and sct, ScT>sCT, sCt, scT and sct, Sct>sCt, and sct) |
| PA | $33.00 \pm 0.82$ | $30.00 \pm 0.71$ | $31.83 \pm 1.36$ | $27.01 \pm 0.91$ | $26.00 \pm 1.07$ | $25.17 \pm 0.83$ | $27.06 \pm 0.81$ | $22.96 \pm 0.73$ | $F = 14.276$, $p < 0.001$ (SCT>Sct, sCT, sCt, scT and sct, SCt>sCt and sct, ScT>sCT and sct, Sct>sct, scT>sct) |
| NA | $18.10 \pm 0.89$ | $16.00 \pm 0.77$ | $20.88 \pm 1.48$ | $16.09 \pm 0.99$ | $23.61 \pm 1.16$ | $21.51 \pm 0.9$ | $25.36 \pm 0.87$ | $20.38 \pm 0.79$ | $F = 11.177$, $p < 0.001$ (SCT<sCT and scT, SCt<sCT, sCt, scT and sct, Sct<sCT, sCt, scT and sct, sCt<scT, scT<sct) |

**Notes.**

SRH, Self-Rated Health; SS, Social Support; SWLS, Satisfaction with Life Scale; PA, Positive Affect; NA, Negative Affect.

SCT (creative), SCt (organized), ScT (fanatical), Sct (autocratic), sCT (moody), sCt (dependent), scT (disorganized), sct (downcast)

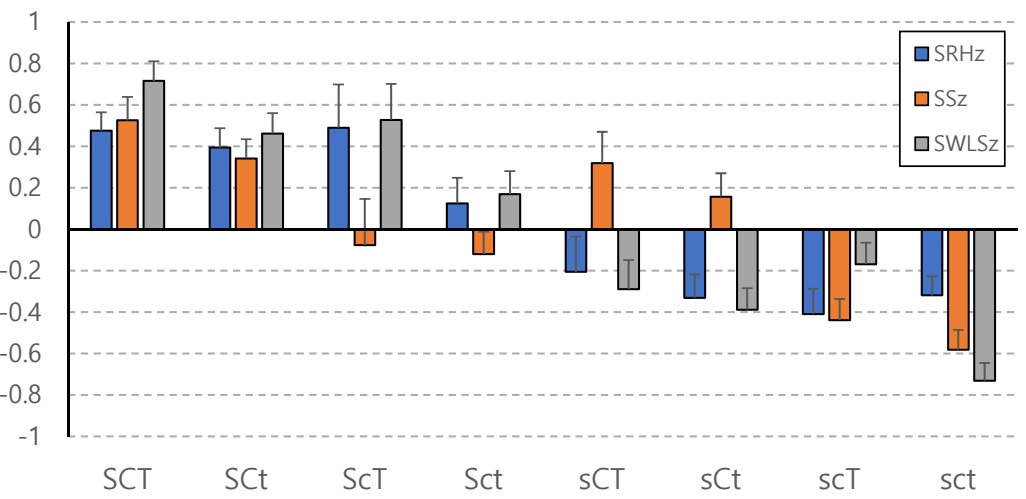

**Figure 1  Well-being measures of character profile groups.**

*Cloninger, 2023*) in various cultural societies regarding the association between character dimensions and multiple aspects of well-being. The results support the hypothesis that character dimensions are distinctively associated with both affective (*e.g.*, positive and negative affect) and non-affective (life satisfaction, subjective health, social support) dimensions of well-being. That is, each character trait independently contributes to well-being, contingent on the influence of other character traits, specifically, through trait by trait interactions.

In the light of the non-affective aspect of well-being, SD had (linear) positive correlations with subjective health, social support and life satisfaction and it had a non-linear positive association with life satisfaction and subjective health in terms of non-affective aspect of well-being. In other words, life satisfaction and subjective health showed positive association with SD. This suggests that considering the interactions among character traits, SD has a clear and consistent impact on life satisfaction and subjective health, which is consistent with previous studies (*Cloninger & Zohar, 2011*; *Giakoumaki et al., 2016*; *Josefsson et al., 2011*; *Moreira et al., 2015*; *Moreira, Inman & Cloninger, 2023*). SD measures a person's being responsible, hopeful, purposeful, and resourceful (*Cloninger, Svrakic & Przybeck, 1993*) and high Self-directedness reported to play a pivotal role in well-being, happiness, or adaptation. Particularly, it was confirmed that the assumption prioritizing SD for psychopathology-free status or well-being (*i.e.*, life satisfaction and subjective health) was repeatedly validated in the present study.

Second, CO had (linear) positive correlations with subjective health, social support and life satisfaction and it had a non-linear positive association with social support in the light of the non-affective aspect of well-being. In other words, social support showed positive association with CO.

That is, social support was strongly associated with higher CO, not with higher SD in non-linear association, which is inconsistent with other previous studies (*Cloninger &*

**A. Composite Health Index of character profile groups**

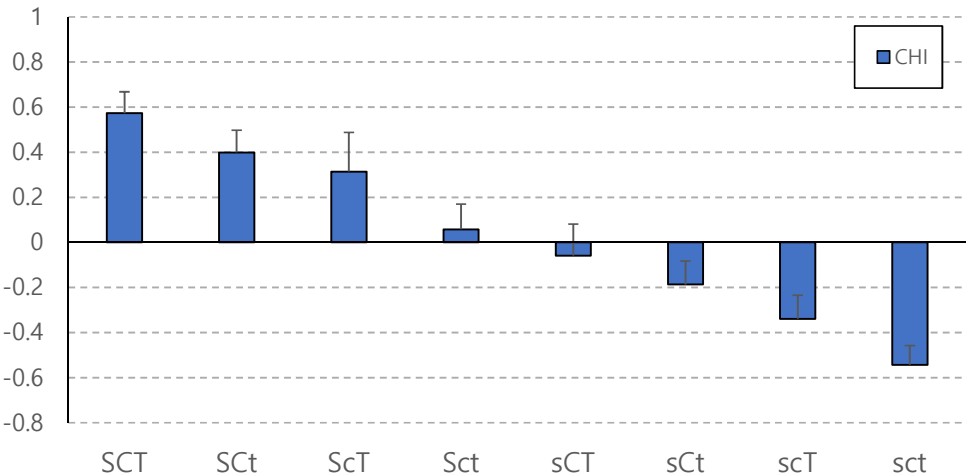

**B. Percentage of people in each character profile with "best health" and "worst ill-health"**

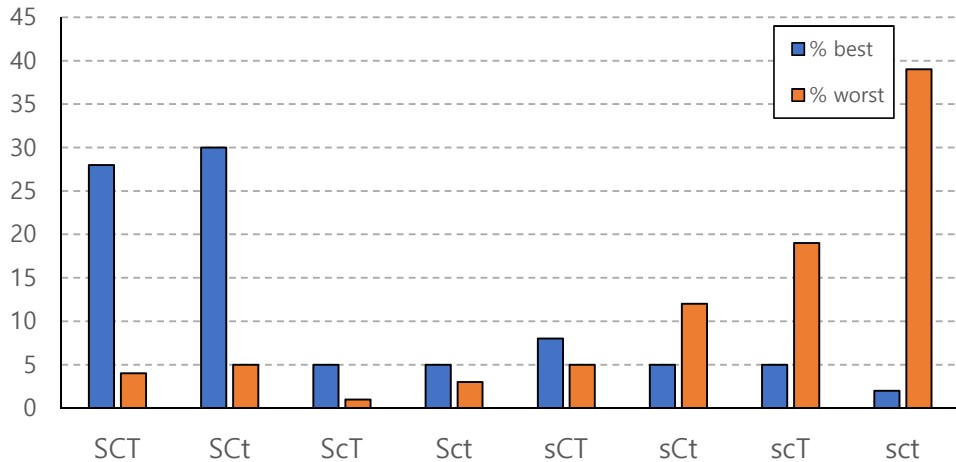

**Figure 2** Composite Health Index of character profile groups.

*Zohar, 2011*; *Josefsson et al., 2011*). Particularly, our results showed that considering the interactions among character traits, CO has a clear and consistent impact on the perception of social support, but SD has not. Cooperativeness measure a person's empathy, helpfulness and social tolerance (*Cloninger, Svrakic & Przybeck, 1993*) and high CO is considered to be associated with social support or acceptance. It implies that CO not SD exerts an important role of social support in Korea society.

In Korean society, individual decision-making and autonomy should be harmonized with those around you and can often be constrained by the rules and expectations of

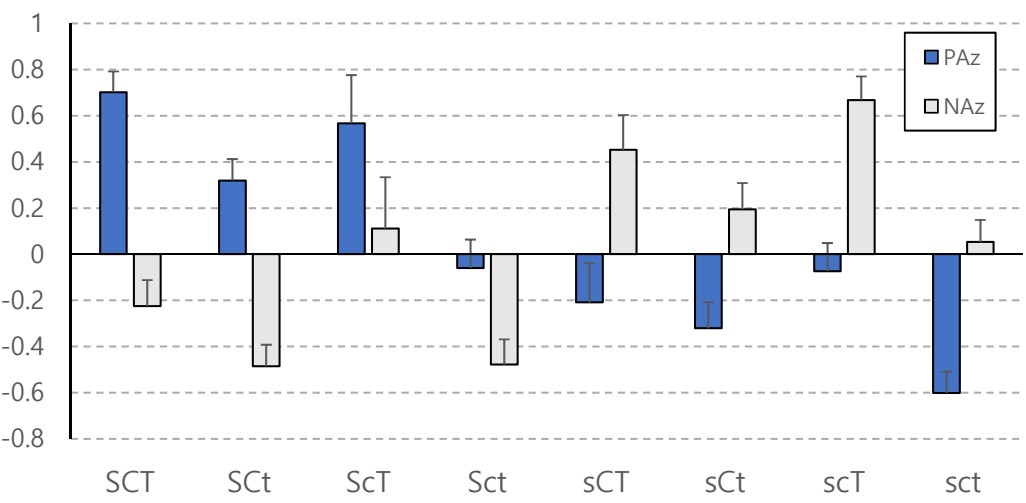

**Figure 3** PA and NA scores of character profile groups.

others. Therefore, the value of CO might take precedence over SD, especially when there is a conflict between the values of individuals and those of others. That is, persons focusing individual goals and purpose might got less experience of social support/acceptance from those around you. Aa saying of "a cornered stone meets the mason's chisel" in Korea could be translated that individuals are advised to act cautiously or maintain a low profile as conspicuous behaviors within a group may be susceptible to criticism or control from the surrounding environment. Collectivism refers to a social, economic, or political ideology that emphasizes the importance of collective action, cooperation, and shared resources over individual interests. It is often contrasted with individualism, which places greater emphasis on independence, personal autonomy, and self-interest. It remains to be explored in subsequent studies whether this reflects the characteristics of East Asian collectivism.

Third, ST did not have (linear) correlations with subjective health and social support but did have a weak correlation with life satisfaction and had no non-linear association with aspect of non-affective aspect of well-being. In other words, ST did not show any meaningful association with non-affective aspect of well-being, which is consistent with other previous studies (*Cloninger & Zohar, 2011*; *Josefsson et al., 2011*; *Moreira et al., 2015*). Meanwhile, in the light of the affective aspect of well-being, it was reported that ST had both positive associations with positive and negative affect in linear and non-linear relationship. This is different from association between SD/CO and affect, which showed the positive relationship with positive affect and negative association with negative affect.

This so-called "twin effect" (*Josefsson et al., 2011*) is consistent with Finnish results in which ST tends to report more positive and negative affect. In all the contrasts when the other character traits are held constant, Self-transcendence increases both negative and positive affect. Not all the differences are significant but the trend is clear. Self-transcendence enhances awareness of connections beyond the individual self with other people and the world as a whole (*Cloninger, Svrakic & Przybeck, 1993*). High ST is often associated with

**Table 5 Comparison between character profiles for perceived health, social support, life satisfaction, positive affect, and negative affect.**

| | | SRH | | SS | | SWLS | | PA | | NA | |
|---|---|---|---|---|---|---|---|---|---|---|---|
| | | $t$ | $p$ | $t$ | $p$ | $t$ | $p$ | $t$ | $p$ | $t$ | $p$ |
| **SD** | | | | | | | | | | | |
| | SCT vs sCT | 3.551 | .001 | 1.104 | .272 | 6.141 | .000 | .4989 | .000 | −3338 | .001 |
| | SCt vs sCt | 4.988 | .000 | 1.271 | .205 | 5.865 | .000 | 4.520 | .000 | −4609 | .000 |
| | ScT vs scT | 3.742 | .000 | 1.678 | .096 | 3.427 | .001 | 3.127 | .002 | −2581 | .011 |
| | Sct vs sct | 2.926 | .004 | 3.127 | .002 | 6.464 | .000 | 3.752 | .000 | −3782 | .000 |
| **CO** | | | | | | | | | | | |
| | SCT vs ScT | −.062 | .951 | 2.616 | .010 | .993 | .323 | .638 | .525 | −1674 | .097 |
| | SCt vs Sct | 1.759 | .081 | 3.152 | .002 | 1.916 | .057 | 2.230 | .027 | −.065 | .948 |
| | sCT vs scT | .983 | .327 | 4.291 | .000 | −.689 | .492 | −.764 | .446 | −1083 | .281 |
| | sCt vs sct | −.084 | .933 | 5.022 | .000 | 2.573 | .011 | 2.252 | .026 | .856 | .393 |
| **ST** | | | | | | | | | | | |
| | SCT vs SCt | .630 | .529 | 1.270 | .206 | 1.811 | .072 | 2.413 | .017 | 2.102 | .037 |
| | ScT vs Sct | 1.579 | .118 | .197 | .845 | 1.764 | .082 | 2.883 | .005 | 3.356 | .001 |
| | sCT vs sCt | .636 | .526 | .867 | .388 | .575 | .566 | 0.699 | .486 | 1.205 | .231 |
| | scT vs sct | −.602 | .548 | 1.014 | .312 | 4.219 | .000 | 3.960 | .000 | 3.926 | .000 |

**Notes.**

SRH, Self-Rated Health; SS, Social Support; SWLS, Satisfaction with Life Scale; PA, Positive Affect; NA, Negative Affect.

SCT (creative), SCt (organized), ScT (fanatical), Sct (autocratic), sCT (moody), sCt (dependent), scT (disorganized), sct (downcast)

exhibiting adaptive qualities when confronted with unavoidable suffering such as diseases and death. However, they are also subject to criticism for harboring naïve and magical thinking and subjective idealism that hinder the acquisition of material wealth and worldly power. Therefore, it is possible to interpret situations (*i.e.,* global environmental crisis) more strongly positively (*i.e.,* such is life and that's perfectly alright) or negatively (*i.e.,* it's doomed and we shall die), in a highly subjective manner, causing more positive or negative emotions. Such tendencies can be understood as emerging from the findings of this study and similar interpretations have been corroborated in previous research (*Josefsson et al., 2011*). Fourth, female exhibited lower SD and higher ST compared to men in this study. Generally, SD and ST show no significant gender differences while CO tends to be higher in female among adults (*Gutierrez-Zotes et al., 2015*). However, the findings in Japan (*Suzuki et al., 2009*; *Takeuchi et al., 2011*) which is similar to our study, showed that female had lower SD and higher ST. South Korea and Japan were reported to share similar Confucian traditions, economic growth, and relatively low gender equality indices (*e.g.,* female empowerment and labor participation) (*Park, 2024*; *Tekin, 2023*). These socio-cultural similarities may result in the lower SD and higher ST of female both in South Korea and Japan. However, China, a communist-socialist country with Confucian tradition demonstrated somewhat inconsistent results with South Korea and Japan: female reported higher SD and lower ST (*Chen, Lu & Kitamura, 2013*; *Wang et al., 2019*). It might reflect higher female empowerment and labor participation compared to South Korea and Japan. This interpretation related to cultural backgrounds is speculative and thus requires further research for validation regarding gender differences related to TCI research.

Fifth, universality of character profiles despite different versions of the TCI across various cultures was revealed in the current study. The most common profiles of SCt (organized, 18.0%) and the sct (depressive, 17.3%) in Korean adult sample were similarly ($\chi^2 =4.82$, $p= .57$) prevalent in Israel (16.8% and 15.7%), Finland (18.8% and 17.0%), and Greece (17.0% and 15.2%) (*Cloninger & Zohar, 2011*; *Giakoumaki et al., 2016*; *Josefsson et al., 2011*) using adult samples. The SCT (13.5%) profile with best health was similarly ($\chi^2 =6.56$, p =.09) represented in Israel (16.0%), Finland (17.8%), and Greece (15.2%) in Greece.

Sixth, the ScT (fanatical or paranoid) and sCT (moody) profiles had the lowest prevalence, at 4.9% and 8.0%, respectively. In similar studies conducted in Finland, Portugal, and Greece (*Giakoumaki et al., 2016*; *Josefsson et al., 2011*; *Moreira et al., 2015*), the fanatical (ScT) profile was found to be the least frequent profile although this profile was not observed in Israel (*Cloninger & Zohar, 2011*). On the other hand, the moody (sCT) profile, one of maladaptive character profiles was the most common in this study, which was only observed in the Israel (*Cloninger & Zohar, 2011*). Therefore, future studies should test the hypothesis that the moody character profile might have culture-bound attributes specific to Asian regions. In addition, a high female proportion of the moody profile was found in the current study, which is also observed in the Finn data (*Josefsson et al., 2011*). The gender ratio should be included in the future analyses.

The current and past research (*Cloninger & Zohar, 2011*; *Giakoumaki et al., 2016*; *Josefsson et al., 2011*; *Moreira et al., 2015*; *Moreira, Inman & Cloninger, 2023*) suggests that well-being is a multidimensional concept that may exhibit significant variations throughout its development. This also implies that each facet of character contributes uniquely to well-being, with its impact strongly influenced by interactions with other character dimensions. The effects of character on well-being are highly non-linear, contingent on specific configurations whose influences vary across distinct aspects of well-being.

In conclusion, the present study shows generally universal but unique association between dimension of character and well-being characteristics. While SD is an important factor in the expression of general well-being (*i.e.,* life satisfaction, subjective health), CO is emerged as an exceeding status over SD in terms of social support. In addition, the twin effect of ST should also be highlighted. However, there is no study to investigate the association between TCI character dimensions and well-being directly using East-Asian sample (*Mitsui et al., 2013*; *Wang et al., 2019*). Therefore, future studies to explore these associations between character and well-being should follow in order to elucidate whether this finding were universal characteristics in East-Asia.

One limitation in our attempt to replicate the study is that different assessment instruments, such as those measuring social support and subjective health, were utilized in other countries. Nevertheless, we employed measures that were closely comparable in various languages, as evidenced by the consistency of most findings in previous research. In the same context, we have very similar findings regarding TCI and its relationship with other measures despite different version of TCI. These results indicate robustness of the biopsychosocial model underlying the TCI as well as other well-being measures. Another limitation in both this study and the original ones is that the observations rely

on a cross-sectional sample, making it challenging to establish causation. Therefore, the connection between personality and perception of well-being should be replicated in a longitudinal approach which might give the results more credibility. In addition, future studies should address the gender difference of character dimensions or profiles for shedding light on the cross-cultural perspectives. Lastly, the sample utilized for this study was drawn from college students. Thus, it is necessary to examine a broader range of age groups, such as adolescents or adults in their 30s or older in order to enhance the generalization of the study.

This study utilizes diverse character profiles to comprehensively describe individual dimensions while acknowledging trait interactions. Employing a person-centered approach, the research provides nuanced insights into human development. Results reveal that non-linear interactions effectively explain well-being, emphasizing the significance of incorporating both linear and non-linear assumptions in models. The findings underscore the impact of each TCI-measured character dimension on affective and non-affective well-being through sociocultural interactions. Future well-being studies should assess multidimensional aspects and their correlation with personality traits across diverse cultural contexts.

### Funding

The authors received no funding for this work.

### Competing Interests
C. Robert Cloninger is an Academic Editor for PeerJ and is the Director of the Anthropedia Institute, a 501(c)(3) nonprofit organization that trains people to become certified professional health and wellness coaches specialized in promoting mental health and well-being.

### Author Contributions
- Soo Jin Lee conceived and designed the experiments, performed the experiments, analyzed the data, prepared figures and/or tables, authored or reviewed drafts of the article, and approved the final draft.
- C. Robert Cloninger conceived and designed the experiments, authored or reviewed drafts of the article, and approved the final draft.
- Han Chae conceived and designed the experiments, performed the experiments, analyzed the data, prepared figures and/or tables, authored or reviewed drafts of the article, and approved the final draft.

### Human Ethics
The following information was supplied relating to ethical approvals (*i.e.,* approving body and any reference numbers):

The study was conducted according to the guidelines of the Declaration of Helsinki, and approved by the Institutional ethic board of Kyungsung University (KSU-19-03-005 [2019.3.29]).

## Data Availability

Raw data is available as Supplemental Information.

## Supplemental Information

Supplemental information for this article can be found online at http://dx.doi.org/10.7717/peerj.18379#supplemental-information.

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
