# Peer review of "The associations between well-being and Cloninger’s personality dimensions in a Korean community sample"

_PeerJ, doi:10.7717/peerj.18379_

## Round 0.1 · original submission · Major Revisions

Please address the Reviewer comments. These will further enhance the submission.

·

Basic reporting

The basic reporting was clear and succinct, the references were relevant to the work presented,

There were no hypotheses presented, rather a general research question was formulated: "we aimed to explore whether the associations between personality and well-being would appear similarly in Western and Eastern cultural context" (lines 130-131).

The internal reliability of the scales of the Korean version of the TCI used in the study are not reported. This makes no sense to me - this is a very minimal way of showing that the questionnaire performed well, and copyright and commercial considerations cannot be the excuse for not doing so If used in scientific research.

The raw data is not raw it is summary data. So it does not conform with Association of Psychological Science standards,

Experimental design

The experimental design is simple and well-suited for a correlational study in the choice of sample size, and questionnaires.

However, the authors make poor use of the fact that this is a fifth replication of a study design, that has been executed with minor variations in four other cultures - Israel, Finland, Greece and Portugal.

Improving the analytical approach might make this paper interesting, and help in making the argument that there is a relationship between the culture in which individuals develop, and the association found between personality and well-being.

Additional analyses that might be performed:
1. Examine the distribution of the character profiles cross-culturally– compare the distribution of these profiles in Korean university students (Table 1) to that found in a general community sample in Israel and Greece, a cohort study in Finland, and adolescents in Portugal. Is the distribution culture bound or is it universal?
2.Use the composite health index (CHI) to examine character profile distribution and conversely use character profiles to examine best and worst CHI (figures 4 and 5 in Cloninger and Zohar 2011).
3. Investigate the sex-differences in character traits (women high in ST and low in SD) relative to men, in the cross-cultural perspective, and include the findings in the discussion,

Validity of the findings

The interpretation of the findings and thus the validity is less convincing than they can be.


If the additional analyses I suggest are performed, the cultural attribution of the findings can be much better validated.

·

Basic reporting

No comment

Experimental design

No comment

Validity of the findings

No comment

Additional comments

Thank you for the opportunity to review the Manuscript “The associations between well-being and Cloninger’s personality dimensions in a Korean community sample (#98256)” for PeerJ.

Understanding the processes underlying well-being is certainly of major importance to science and societies. Therefore, studies that contributes to build knowledge on communalities and specificities of the expression of well-being in different cultures are of major relevance.
This is the case of the present manuscript. Major strengths of this manuscript including describing the associations between personality and well-being 1) using a biopsychobiological theory of personality, 2) considering the non-linear dynamics among different process within the individual, 3) using analytical methodologies that allow for a comparison between the results of the present study and the results of other studies addressing the same question and, 4) in a Asian (Korean) sample. So, I applaud the good job that the authors made in the study described by this manuscript.
I was particularly interested in reading the manuscript when I realized that the authors conducted this study using the Short Version of the TCI-R. And it is very interesting that the results – including those from non-linear associations – are so consistent with the results with other studies that used different versions of the TCI-R, either the 240 items version or the Junior TCI-R.
In my opinion, I think that this is a relevant question, because we know that often changes in the characteristics of versions of a questionnaire (reduction of items, for example) result in marked differences in the various validity indicators of the instrument. This is particularly relevant in the cases of non-linear analysis, such as it is the case.
Because this is an indicator of the robustness of the theory underlying the assessment instrument, I wonder if the authors could make this strength of the manuscript explicit on the text, so the readers can be informed also about this.

---

## Round 0.2 · Minor Revisions

Please highlight the remaining issues within the manuscript.

·

Basic reporting

This is a very significant revision. Well done. The additional analyses make it much more interesting and I expect the paper in its next form to appeal to a much wider audience.
The language needs editing by a native speaker - there are countless linguistic errors which need correction.
One of the comments the authors did not address was the universality of the distribution of character profiles, comparing the distribution in Korea, Israel and Finland (and if they can find them in Greece and Portugal). I just eyed the (now) published numbers for the high character SCT profile:
In Korean students 13.5%
In Finland 16%
and in Israel 18%
They seem very similar but showing this statistically (by chi-square) would make the point, and suggest that high character is similarly represented in very different cultures; as measured by different versions of the TCI; with different sampling designs, enhancing the universality of character profiles across cultures.

Experimental design

very suitable

Validity of the findings

convincing

Additional comments

No additional comments - just correct the language please and check the distribution of profiles

·

Basic reporting

no comment

Experimental design

no comment

Validity of the findings

no comment

Additional comments

no comment

---

## Round 0.3 · accepted · Accept

Following the review process this paper is now of an acceptable standard.

·

Basic reporting

This paper reads very well and will be very interesting, Great job!

Experimental design

appropriate

Validity of the findings

well founded

Additional comments

None